# Jump Self-attention: Capturing High-order Statistics in Transformers

**Haoyi Zhou**
BDBC
Beihang University
Beijing, China 100191
haoyi@buaa.edu.cn

**Siyang Xiao**
BDBC
Beihang University
Beijing, China 100191
xiaosy@act.buaa.edu.cn

**Shanghang Zhang**
School of Computer Science
Peking University
Beijing, China 100871
shanghang@pku.edu.cn

**Jieqi Peng**
BDBC
Beihang University
Beijing, China 100191
pengjq@act.buaa.edu.cn

**Shuai Zhang**
BDBC
Beihang University
Beijing, China 100191
zhangs@act.buaa.edu.cn

**Jianxin Li**[*]
BDBC[†]
Beihang University
Beijing, China 100191
lijx@buaa.edu.cn

## Abstract

The recent success of Transformer has benefited many real-world applications, with its capability of building long dependency through pairwise dot-products. However, the strong assumption that elements are directly attentive to each other limits the performance of tasks with high-order dependencies such as natural language understanding and Image captioning. To solve such problems, we are the first to define the Jump Self-attention (JAT) to build Transformers. Inspired by the pieces moving of English Draughts, we introduce the spectral convolutional technique to calculate JAT on the dot-product feature map. This technique allows JAT's propagation in each self-attention head and is interchangeable with the canonical self-attention. We further develop the higher-order variants under the multi-hop assumption to increase the generality. Moreover, the proposed architecture is compatible with the pre-trained models. With extensive experiments, we empirically show that our methods significantly increase the performance on ten different tasks.

## 1 Introduction

Transformers have made great success in many domains, such as NLP, Vision, Time-series, and Speech. However, the giant model GPT-3 [2] and switch Transformer [7], with their massive layers and over-parameterized architectures, have not further achieved significant performance improvements since the BERT model [5] beat the Human Performance in GLUE [21] and SQuAD [12]. Recently two visualization works [19, 3] reveal that the self-attention mechanism mainly learns superficial connections among inputs. The principle of direct accessing among inputs in the dot-product self-attention lies as the core of above models, which solely relies on their mutual similarity in the vector space. However, as "every coin has two sides", while the dot-product is simple and effective in capturing long-range dependencies, it has limited capability in learning high-order dependencies.

To solve this problem, we are inspired by the pieces moving of English Draughts, a checker game spread to Great Britain in about 1100. It allows two moving rules: one is the *simple move* from the

---

[*]Jainxin Li is the corresponding author.

[†]BDBC is the abbreviation for the Beijing Advanced Innovation Center for Big Data and Brain Computing.

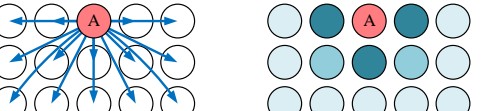 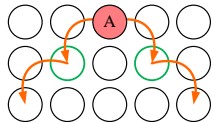 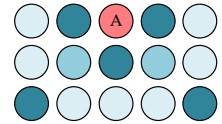

(a) Dot-product attention with direct connections.      (b) High-order attention with "jump" connections.

Figure 1: **(a) The dot-product self-attention:** Left side: The direct connections (the blue lines with arrows) are established between the input A (the red node) and other inputs via the dot-product self-attention. Right side: The nodes with different shades of blue represent the corresponding attention of the feature map. **(b) The high-order self-attention:** Left side: We build "jump" connections from input A (the red node) to previous unattended nodes at the corner, which is conditioned on the direct connections on "stepping stones" (the green circled nodes). Thus the dot-product feature map is enhanced with high-order connections as shown on the right side.

current position to an adjacent position; and the other is the *capture move* jumping over the "stepping stones" (opponent's pieces) from front to back. Multiple jumps are possible and forced in this game. As shown in Fig.(1), we can employ the high-order self-attention connections, like the *capture moves* in English Draughts, to enhance the feature map of dot-product self-attention. It helps to align high-order global dependencies in Transformer models. As a more concrete example in natural language understanding, we are given two sentences "Book costs more than pencil." and "Book costs less than computer". "Computer" never attends to "pencil" until they were both compared with "Book", thus "Book" is the "stepping stones" in creating high-order attention between "computer" and "pencil" in Fig(1b). More specifically, they forms a soft triadic closure. We will discuss this example in detail in Section 3.1 and clarify the definition of Jump Self-attention.

In this work, we find that the high-order self-attention exists in the dot-product computation but it is too weak to contribute much to the attention layer's output. Only a few heads in Transformers can capture such high-order dependencies, which is shown in Fig.(2) with further discussions. Instead of finding such "miracle" heads, we reformulate them to enhance the high-order dependencies with the feature propagation technique in graph learning. Meanwhile, we build an efficient variant in computing-sensitive situations based on the sparsity measurement [25], and build higher-order variants in comprehensive situations using MixHop [1].

To the best of our knowledge, we are the first to define the _Jump Self-attention_ (JAT) in Transformer models, where we formulate the high-order statistics calculation into a graph propagation problem. It can be used interchangeably with the canonical self-attention [18]. The JAT-enhanced Transformer allows for fine-tuning on the popular pre-trained models and achieves superior performance on various tasks, such as Natural Language Understanding and Question Answering. Our main contributions can be summarized as:

- We discover the existence of JAT in the Transformers' dot-product computation and derive its formal definition to facilitate further analysis.
- We propose JAT through the jump aggregation and high-order dependencies' propagation without additional network parameters, and the efficient variant alleviates the extra computation cost.
- We propose the higher-order variants of JAT and masking strategy to allow its interchangeable usage with the canonical self-attention on the pre-trained models.
- We conduct experiments on fifteen different tasks to examine the effectiveness of the proposed methods. Results demonstrate the success of JAT in achieving state-of-the-art performance.

## 2 Preliminary

The self-attention was introduced by Transformer [18] model for the encoder-decoder architecture in NLP to build long-range dependencies from a variable-length source sentence. The alignment ability of self-attention to learn to acquire direct access between important items has made it an efficient component in many applications [5, 11, 25, 23]. For an input $\mathbf{X} \in \mathbb{R}^{L \times d}$, we define the single-headed self-attention in the following form:

$$\text{Attention}(\mathbf{Q}, \mathbf{K}, \mathbf{V}) = \text{softmax}\left(\frac{\mathbf{Q}\mathbf{K}^\top}{\sqrt{d}}\right)\mathbf{V} \quad, \tag{1}$$

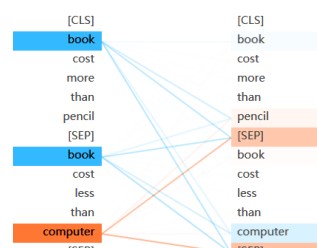 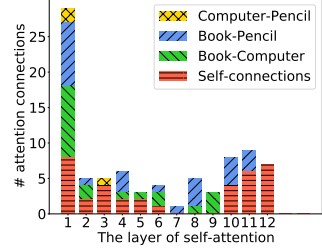 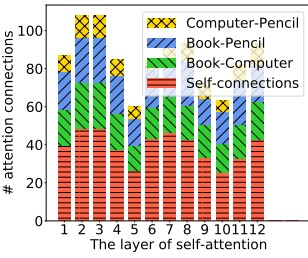

(a) AttnVis {layer7, head5}.     (b) Significant connections (15.2%).     (c) Major connections (70.5%).

Figure 2: A case study of sentence "Book cost more than pencil. Book cost less than computer." on the pre-trained BERT model [5]. **(a) Attention Visualization:** We specifically select the third layer and draw the attention score of two words "book" and one word "computer" at the fifth head in the multi-head self-attention. It is an extraordinary example to show that the word "computer" in the second sentence has established a moderate connection with the word "pencil" in the first sentence. The visualization tool is available online[3]. **(b) Statistic of self-attention for significant connections across all heads:** We first calculate the mean value and standard deviation over all the pairwise self-attention scores. And we select the 15.2% of the self-attention connections among the three words "book", "pencil", and "computer" that their scores are greater than the threshold (mean + 0.5 std). The previous example in (a) is the miracle attention connection at the third layer, and it works by acquiring more connections than other layers. **(c) Statistic of self-attention for major connections across all heads:** We lower the threshold to (mean - 0.5 std) and it covers 70.5% self-attention connections. It shows that the dot-product self-attention attempts to build the "computer-pencil" connections, but they are a few weak ones and contribute little to layer output.

where the layer output is the linear combination of values $\mathbf{V}$ weighted by the attention scores between queries $\mathbf{Q}$ and keys $\mathbf{K}$. Note that the queries, keys and values are projected by different linear transformations $\{\mathbf{W}_Q, \mathbf{W}_K, \mathbf{W}_V\}$ respectively. In practice, multiple attention heads are used to learn robust patterns [3, 19] among different transformations. It divides the inputs into $H$ groups and computes the single-head attention on each group separately. We concatenate the feature map of all groups as the attention layer output.

## 3 Methods

In this section, we first introduce the necessity of acquiring the high-order connections in the Transformer model. Then, we describe the proposed Jump Self-attention.

### 3.1 What is the *High-order Statistics*?

The BERT [5] model and its pre-trained successors utilize the self-attention, like GPT-3 [2] and switch Transformer [7], and they achieve distinct performance improvements on various NLP tasks. However, what the self-attention mechanism has learned remains to be a challenging problem for architecture optimization in Transformer. Recent visualization techniques on self-attention mechanism have shed light on the further explanation for this problem. In the BERT model, Vig [19] pointed out that different heads of the same layer highlight coarse positional patterns, such as 'CLS' or 'SEP' tokens, more frequently than lexical patterns, even though they are trained with separate random projections. Clark et al. [3] claimed that BERT's attention heads often exhibit similar behaviors within the same layer, such as attending to delimiter and positional tokens and broadly attending the entire sentence, whereas a few deviating heads hold an eye on linguistic pairs. Due to the space limit, a more comprehensive related work will be provided in the Appendix A. This paper investigates whether it is worth enhancing lexical patterns in self-attention.

To motivate our approach, we perform an individual case study on investigating the lexical patterns of self-attention heads. Based on the pre-trained model on BERT [5], we consider logical reasoning questions on two continuous sentences "Book cost more than pencil" and "Book cost less than computer". It seeks to find the price between entities that have not been compared, like "pencil"

---

[3]The tool can be acquired at `https://github.com/jessevig/bertviz`.

and "computer". Note that the articles are omitted to avoid interference. In Fig.(2a), we present the visualization of three words' self-attention scores at the third layer and fifth head. The darker line color denotes a higher self-attention score, namely stronger connections. The word "book" shows strong attention connections (deeper blue lines) to the positional token 'SEP' in the first and second sentences, meanwhile, they also attend to the entity words "pencil" and "computer" (lighter blue lines). While the attention scores between the word "computer" and token 'SEP' are striking, we note that the word "computer" builds a mild attention connection to the word "pencil" crossing two sentences. From the entity extraction perspective, the mutual attention connections among three words "book", "pencil" and "computer" seem trivial.

However, the concrete finding in Fig.(2b) undermines this emotional thinking and it reveals the potential value of attention connections between "computer" and "pencil". We first calculate the mean value and standard deviation over all the pairwise self-attention scores. And we select the attention connections that their scores are greater than the threshold (mean + 0.5 std). We only draw the local connections among the three words and it covers 15.2% of the total as the significant local connections. The stronger "Computer-Pencil" attention connection, as it in Fig(2b), is the rare self-attention computation that happens at the third layer, which distinguishes it from the trivial "Book-pencil" and "Book-Computer" attention connections. As we discussed above, there are many connections from three words to the delimiter 'CLS' and positional tokens 'SEP' but they did not form the "Computer-Pencil" connection. Furthermore, in Fig.(2b), there are many "Book-Pencil" connections and "Book-Computer" connections that exist in other layers but they all fail to capture the "Computer-Pencil" connection. The fifth head in the third layer is the miracle head that can establish the strong linguistic connection between "computer" and "Pencil" and it forms lexical self-attention patterns across different sentences. We call this an exact example of the *High-order Statistics*, and it is formulated on input pairs which satisfy the following definition:

**Definition 3.1.** $(\delta, p, \mathbf{W}_Q, \mathbf{W}_K)$-*similar* For vector inputs $\mathbf{x}_i$, $\mathbf{x}_j$ and $\mathbf{x}_k$, there exists a single head self-attention with query transformations $\mathbf{W}_Q$ and key transformations $\mathbf{W}_K$, and the self-attention computation forms the feature map score as $\mathbf{S}_{ij} = (\mathbf{x}_i \mathbf{W}_Q)(\mathbf{x}_j \mathbf{W}_K)^\top$ such that, under the $\delta$-significant threshold $p$,

$$if\ \mathbf{S}_{ik} \geq p\ and\ \mathbf{S}_{jk} \geq p,\ then\ \Pr[\mathbf{S}_{ij} \geq p] > \delta$$
$$otherwise,\ then\ \Pr[\mathbf{S}_{ij} \geq p] \leq 1 - \delta$$

Thus the inputs $\mathbf{x}_i$ and $\mathbf{x}_j$ are called $(\delta, p, \mathbf{W}_Q, \mathbf{W}_K)$-similar w.r.t the input $\mathbf{x}_k$.

In Fig.(2c), we relax the restriction on significant connections and select the major connections covering 70.5% local connections among the three words. As a natural result of decreasing the significance $\delta$ and threshold $p$ in Definition 3.1, it emerges more high-order self-attentions, i.e., connections between "Computer" and "Pencil". This implies that weak high-order self-attentions exist in the canonical self-attention but these connections are too delicate to contribute to self-attention's layer out.

## 3.2 Calculate the *Jump Self-attention*

The Jump Self-attention (JAT) is based on the vanilla Transformer architecture to be consistent with the latest improvement on it. Inspired by the pieces moving in Fig.(1), we build a transaction graph on the self-attention feature map. Then JAT utilizes the GCN operator to allow "jump" connections to traverse through different queries and key pairs, and the weak connections under the similar condition are enhanced to help enforces the desired triadic closure property.

Let the attention score of single-headed self-attention in Eq.(1) be $\mathbf{S} = \mathbf{Q}\mathbf{K}^\top$, we notice that the $\mathrm{softmax}(\cdot)$ function on $\mathbf{S}/\sqrt{d}$ is row-wise, which makes the self-attention layer output the correct re-representation of value $\mathbf{V}$. To understand the underlying meaning of high-order statistics in attention computation, we can take a column view of $\mathbf{S}$ in Fig.(3a). Different from the row view where each row reflects one query's scores on all keys, in the column view, each key column represents a similarity measurement of all queries against itself.

Consider an intuitive example from the previous case study, we draw the attention feature map with five queries and keys in Fig.(3b). We firstly present the dot-product self-attention scores on the left side. The self-attention scores $\mathbf{S}_{31} = \mathbf{Q}_3(\mathbf{K}_1)^\top$ and $\mathbf{S}_{41} = \mathbf{Q}_4(\mathbf{K}_1)^\top$ highlight in the column vector of the first key, as we can see the crosshatch under the column view (the red shade), which indicates that inputs $\mathbf{x}_3$ and $\mathbf{x}_4$ attend to the input $\mathbf{x}_1$ from the $\mathbf{K}_1$'s observation. The connection between $\mathbf{x}_3$

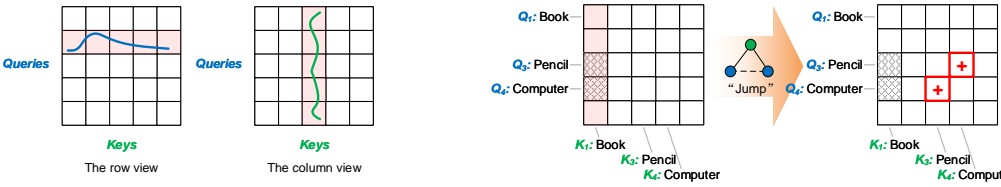

(a) The different views.

(b) The overview of the high-order self-attention.

Figure 3: **(a) Two different views on self-attention scores:** We present the row view of self-attention scores at the top and the column view at the bottom. The solid blue line denotes the second query's self-attention distribution over all the keys, and the single apex indicates focused attention. The solid green line represents the self-attention distribution between all queries and one key, and the multi-peak reveals three stronger attention connections for that key. **(b) An example of finding the "jump" connection:** We can utilize the "Pencil" and "Computer" attention to "Book" to enhance a closer similarity between them, namely, from the crosshatch to the red marked squares.

and $\mathbf{x}_4$ are weak (the blank square). Based on the high-order statistics, we have a high probability $\delta$ that the inputs $\mathbf{x}_3$ "*Pencil*" and $\mathbf{x}_4$ "*Computer*" becomes $(\delta, p, \mathbf{W}_Q, \mathbf{W}_K)$-similar w.r.t the input $\mathbf{x}_1$ "*Book*". But the dot-product self-attention can not capture this kind of similarity for $\mathbf{x}_3$ and $\mathbf{x}_4$. If we could modify the linear transformation $\mathbf{W}_Q$ and $\mathbf{W}_K$ such that the red marked square attains larger attention scores while others maintains, and the attention score between $\mathbf{x}_3$ and $\mathbf{x}_4$ grows larger enough than the significance threshold $p$. We call this attention adjustment a successful "jump", and it could happen for the individual key.

We assume that there exists a "miracle head", whose self-attention feature map contains both dot-product similarity and high-order statistics. Instead of modifying the linear transformation $\mathbf{W}_Q$ and $\mathbf{W}_K$, we can direct "correct" the self-attention feature map by running thorough all triplets and adjusting the scores by specific keys. However, this direct operation will introduce an extra computation of attention combination $\mathcal{O}(L^3)$ w.r.t the input length $L$. And it is hard to acquire a back-propagation friendly implementation.

To formulate the modification of feature map, we relax the objective of finding the miracle heads to enhance the weak "jump" connection in canonical self-attention computation. Under the row view, we build a graph $G = (V, E)$ from the self-attention scores, where the queries compose the node set $V$ with the row vector of $\mathbf{S}$ as node features and each edge is defined by a adjacency matrix $\mathbf{A}$. Then, we introduce $\mathbf{A}$'s computation. Subscribing the matrix with $i^\top$ returns the $i$-th column as a vector. We select the $j$-th key as the current column, then the sub-adjacency matrix under the $\mathbf{K}_j$ is defined:

$$\mathbf{A}^{(\mathbf{K}_j)} = [\mathbf{U}_j - \mathrm{diag}(\mathbf{U})_j]_\rho, \qquad \text{where} \quad \mathbf{U}_j = \frac{\mathbf{S}_{j^\top} \otimes \mathbf{S}_{j^\top}}{d} \quad . \tag{2}$$

The operation $\otimes$ denotes the tensor product and we use the similar $d$-dimension normalization in the canonical self-attention [18]. $\mathbf{U}_j$ is a symmetric matrix containing input pairs with $(\delta, p, \mathbf{W}_Q, \mathbf{W}_K)$-similarity, like $\mathbf{x}_3$ and $\mathbf{x}_4$ in the previous example. $[\cdot]_\rho$ is an indicator function for each matrix item that returns $1$ if the item is greater than the $\rho$, otherwise return $0$. Here, we break the "both" requirements in Definition 3.1. Thus, the threshold $\rho$ becomes a hyperparameter reflecting the magnitude of $\delta$ and $p$. We go through all keys, the $j$-th sub-adjacency matrix reflects the similarity between different queries w.r.t $\mathbf{K}_j$.

Inspired by the layer aggregation [17, 10], the *summation* of weighted adjacency matrix yield larger gap between isolated values and bulk eigenvalues, which has been widely used in helping the community detection task [14, 4]. The sub-adjacency matrices are describing the connections between all the queries, i.e, different layers in a multiplex network, we can also leverage the same operation for a better edge probability recovery – similarity detection. Then, we can add up all the sub-adjacency matrix into the weighted one:

$$\mathbf{A} = \frac{1}{L} \sum_j \mathbf{A}^{(\mathbf{K}_j)} \quad . \tag{3}$$

Our next question is to enhance the "jump" connection for individual inputs. The Graph Convolutional Neural network (GCN) could perform local aggregate operators over the graph adjacency matrix,

which is proposed by Scarselli et al. [15] and is further simplified by Duvenaud et al. [6] and Sukhbaatar et al. [16]. We employ it as a specific architecture for encouraging the attended items to become similar in both queries and keys. Then we have:

$$\mathbf{Q}^{'} = \hat{\mathbf{A}}\mathbf{Q}\mathcal{W}_Q, \quad \mathbf{K}^{'} = \hat{\mathbf{A}}\mathbf{K}\mathcal{W}_K \tag{4}$$

where $\hat{\mathbf{A}} = \tilde{\mathbf{D}}^{-\frac{1}{2}}\tilde{\mathbf{A}}\tilde{\mathbf{D}}^{-\frac{1}{2}}$, $\tilde{\mathbf{A}} = \mathbf{A} + \mathbf{I}$ and $\tilde{\mathbf{D}}$ is a diagonal matrix from the column-sum of $\mathbf{A}$, where every node can only attend to its nearest neighborhood with the learnable graph filter $\mathcal{W}$. The GCN will make the selected queries/keys propagate its features, which causes a higher score when encountering each other in self-attention computation. In other words, GCN encourages high-order self-attention's propagation through individual keys' similarity measurement. Thanks to this property, the self-attention score will be updated as:

$$\mathbf{S}^{'} = \mathbf{Q}^{'}\mathbf{K}^{'\top} = \hat{\mathbf{A}}\mathbf{X}\mathbf{W}_Q\mathcal{W}_Q\mathcal{W}_K^{\top}\mathbf{W}_K^{\top}\mathbf{X}^{\top}\hat{\mathbf{A}}^{\top} = \hat{\mathbf{A}}\mathbf{X}(\mathbf{W}_Q\mathcal{W}_Q)(\mathbf{W}_K\mathcal{W}_K)^{\top}\mathbf{X}^{\top}\hat{\mathbf{A}}^{\top} \quad . \tag{5}$$

If we aggregate the graph filter $\mathcal{W}$ into the random projection $\mathbf{W}$ as a new projection $\mathbf{W}^{'}$, the score $\mathbf{S}^{'}$ becomes $\hat{\mathbf{A}}(\mathbf{X}\mathbf{W}_Q^{'})(\mathbf{X}\mathbf{W}_K^{'})^{\top}\hat{\mathbf{A}}^{\top}$. This could be consider as a way to modify the original learnable matrix $\mathbf{W}_Q$ into the new projection $\mathbf{W}_Q^{'}$. Note that the proposed method maintains the same parameter scale. Immediately, we could derive the self-attention score after applying the GCN operator on both queries and keys, namely the "jump" operation:

$$\Phi(\mathbf{S}) = \hat{\mathbf{A}}\mathbf{S}\hat{\mathbf{A}}^{\top} \quad , \tag{6}$$

which forms an approximated graph operator repeating propagation over the graph to enforce the desired triadic closure, like SGC [22]. The original GCN activation operation is omitted to avoid interfering with later self-attention calculations, which is merged into the following Feed-Forward Net. In this way, we are able to propose the Jump Self-attention:

$$\mathcal{A}(\mathbf{Q}, \mathbf{K}, \mathbf{V}) = \text{softmax}\left(\frac{\Phi(\mathbf{Q}\mathbf{K}^{\top})}{\sqrt{d}}\right)\mathbf{V} \quad . \tag{7}$$

To better take advantage of the pre-trained model, we suggest to split H heads in canonical self-attention into two groups of $H_c$ heads and $H_j$ heads. The $H_c$ heads perform the canonical self-attention operation and the $H_j$ heads perform jump self-attention, then we concatenate their results as the complete layer output. The use of high-order self-attention enables acquiring high-order dependencies over the superficial ones. Overview of JAT's whole process is given in Appendix. The JAT and canonical self-attention can be used interchangeably in Transformer models.

### 3.2.1 Masking tricks

The mask of self-attention [18] plays an important role in avoiding the auto-regressive during the inference phase. The GCN operations may cause leftward information to leak between queries. If JAT is applied in the decoder, we have to prevent this in building the adjacency matrix, where we reduce the edges from afterward queries to previous queries. We implement this inside of JAT by forcing the upper triangular of adjacency matrix to 0 in Eq.(3). The canonical self-attention's masking operation on the feature map is preserved after applying GCN in Eq.(5) such that we can eschew the information leakage in $\text{softmax}(\cdot)$ function.

### 3.2.2 Efficient variants

The computation complexity of JAT remains $\mathcal{O}(L^3)$ w.r.t the input length $L$, because we have to traverse all keys in the adjacency matrix aggregation of Eq.(3). We previously designed a sparsity measurement [25] to find the most significant queries in self-attention, which motivates us to employ it in selecting the primary keys:

$$M^{(\mathbf{K}_j)}(\mathbf{S}) = \max_i(\mathbf{S}_{j\top}) - \max_i(\mathbf{S}_{j\top}) \quad . \tag{8}$$

We follow the sampling strategy and choose Top-$u$ Keys. Thus, the original self-attention $\mathbf{S}$ reduces to $\bar{\mathbf{S}} \in \mathbb{R}^{L \times u}$ and we acquire the efficient variant JAT$^{\dagger}$ in the $\mathcal{O}(L^2)$. Recalling that the JAT shares the same parameter scale, and the JAT$^{\dagger}$ holds the same computation complexity when compared with the canonical self-attention.

Table 1: The performance comparison on the GLUE benchmark (dev sets).

| Model | CoLA 8.5k | MRPC 3.5k | RTE 2.5k | STS-B 5.7k | QNLI 108k | QQP 363k | SST-2 67k | WNLI 0.64k | MNLI 392k | Average - |
|---|---|---|---|---|---|---|---|---|---|---|
| ELMo | 44.1 | 76.6 | 53.4 | 70.4 | 71.1 | 86.2 | 91.5 | 56.3 | 68.6 | 68.7 |
| BERT$_{base}$ | 56.3 | 88.6 | 69.3 | 89.0 | 91.9 | 89.6 | 92.7 | 53.5 | 86.7 | 79.7 |
| BERT-$A^3$ | **62.3** | **90.2** | **73.9** | 90.1 | 91.1 | 90.6 | 92.9 | 55.6 | **87.3** | 81.5 |
| BERT-JAT (ours) | 61.6 | 89.1 | 73.3 | **90.5** | **93.2** | **91.6** | **93.5** | 57.3 | 85.6 | **81.7** |
| BERT-JAT$^\dagger$ (ours) | 61.7 | 89.6 | 72.8 | 90.4 | 92.9 | 91.3 | 93.3 | 56.8 | 85.5 | 81.6 |
| RoBERTa$_{base}$ | 63.6 | 90.2 | 78.7 | 91.2 | 92.8 | **91.9** | 94.8 | - | 87.6 | 86.4 |
| RoBERTa-JAT (ours) | 65.9 | 91.4 | 80.2 | 91.3 | **93.2** | 91.7 | **95.4** | - | **87.7** | **87.1** |
| RoBERTa-JAT$^\dagger$ (ours) | **66.7** | **91.7** | **80.9** | **92.3** | 92.9 | 91.7 | 94.9 | - | 87.5 | **87.1** |

$^1$ JAT$^\dagger$ represents the efficient variant of JAT. And the '-' indicates abandoned experiments.

### 3.2.3 Higher-order variants

Without loss of generality, we can extend the *similar* in Definition 3.1 to a higher order case, where the similarity depends on more than one input. Instead of performing analogical derivations, we can introduce the Mix-hop [1] GCN to capture the multi-hop propagation on the graph of queries. If we refer the canonical self-attention [18] to the first order self-attention and our JAT to the second order one, then we can define the $j$-th order self-attention as:

$$\Phi^j(\mathbf{S}) = \begin{cases} \mathbf{S} & , \text{ if } j = 1 \\ \hat{\mathbf{A}}^{j-1}\mathbf{S}(\hat{\mathbf{A}}^{j-1})^\top & , \text{ if } j > 1 \end{cases} \tag{9}$$

where $\hat{\mathbf{A}}^{j-1}$ represents the adjacency matrix $\hat{\mathbf{A}}$ multiplied by itself $(j-1)$ times. It's trivial to substitute the $\Phi^j(\mathbf{S})$ in Eq.(7) for higher-order self-attention variants $\mathcal{A}^2$, $\mathcal{A}^3$ and etc. We can split the heads into more groups when concatenating multiple self-attentions with different orders.

## 4 Experiments

In this section, we empirically demonstrate JAT's effectiveness on the pre-trained BERT$_{base}$ model with 12 layers, and perform BERT-JAT fine-tuning and evaluation on ten NLP tasks.

### 4.1 Setup

**(a) GLUE.** We conduct JAT's experiments of the language understanding and generalization capabilities on the General Language Understanding Evaluation (GLUE) benchmark [21], a collection of diverse natural language understanding tasks. We also perform additional experiments on SuperGLUE benchmark [20]. **Settings:** We use a batch size of 32 and fine-tune for 5 epochs over the data for nine GLUE tasks. The threshold $\rho$ is selected from $\{4.0, 4.5, \ldots, 10.0\}$. The other settings follow the recommendation of the original paper. Since the proposed JAT can be used interchangeably with canonical self-attention, we perform a grid search on the layer replacement. There are two sets of layer deployment, where the first combination is chosen from $\{\text{Layer}_{1-4},$ $\text{Layer}_{5-8}, \text{Layer}_{9-12}\}$ and the alternative is $\{\text{Layer}_{1-6}, \text{Layer}_{7-12}\}$. Another important selection is the multi-heads grouping, we employ the "side-by-side" strategy as replacing the heads $\{2, 4, 6, 8, 10\}$ with JAT. And we do not use any ensembling strategy or multi-tasking scheme in this fine-tuning. The evaluation is performed on the Dev set. **Metric:** We use three different evaluation metrics on the 9 tasks. Matthews correlation coefficient ($MCC = \frac{TP \times TN - FP \times FN}{\sqrt{(TP+FP)(TP+FN)(TN+FP)(TN+FN)}}$) is used for CoLA, Pearson ($PCC = \frac{N \sum y_i \hat{y}_i - \sum y_i \sum \hat{y}_i}{\sqrt{N \sum y_i^2 - (\sum y_i)^2}\sqrt{N \sum \hat{y}_i^2 - (\sum \hat{y}_i)^2}}$) is used for STS-B, and *accuracy* is used for others. **Platform:** Intel Xeon 3.2GHz + The Nvidia V100 GPU (32 GB) X 4. The code is available at `https://github.com/zhouhaoyi/JAT2022`.

**(b) SQuAD.** We also evaluate on the SQuAD v1.1/v2.0 [12, 13]. The settings is in Appendix D.

### 4.2 Main results

The result of the GLUE experiment is summarized in Table 1. We compare ELMo [8] the baseline of GLUE, BERT$_{base}$ [5], and RoBERTa [9], the pre-trained model where we build JAT varints. Our model BERT/RoBERTa-JAT outperform base models on eight tasks without introducing more

Table 2: The performance comparison on the SuperGLUE benchmark (dev sets).

| Model | CB Acc/F1 | BoolQ Acc | COPA Acc | MultiRC F1/EM | WiC Acc | WSC Acc | RTE Acc | Average |
|---|---|---|---|---|---|---|---|---|
| CBOW | 71.4/49.6 | 62.4 | 63.0 | 20.3/0.3 | 55.3 | 61.5 | 54.2 | 55.4 |
| M.F.C | 50.0/22.2 | 62.2 | 55.0 | 59.9/0.8 | 50.0 | 63.5 | 52.7 | 56.2 |
| $BERT_{base}$ | 94.6/93.7 | 77.7 | 69.0 | 70.5/24.7 | **74.9** | **68.3** | 75.8 | 75.8 |
| $RoBERTa_{base}$ | 92.8/93.7 | 81.5 | 74.0 | 70.7/28.4 | 69.1 | 64.4 | 78.7 | 75.9 |
| RoBERTa-JAT | × | × | **82.0** | × | 70.2 | 65.4 | 80.2 | - |
| RoBERTa-JAT† | **98.2/98.5** | **82.3** | 79.0 | **71.6/29.5** | 70.5 | 67.3 | **80.9** | **78.5** |

¹ JAT† represents the efficient variant of JAT.
² The '-' indicates abandoned experiments, and '×' happens when reaching out-of-memory.

Table 3: The performance on the SQuAD.

| Model | SQuAD v1.1 | | SQuAD v2.0 | |
|---|---|---|---|---|
| | EM | F1 | EM | F1 |
| BiDAF-ELMo | - | 85.6 | 63.4 | 66.2 |
| $XLNet_{base}$ | 89.7 | 95.1 | 87.9 | 90.6 |
| $BERT_{base}$ | 81.2 | 87.9 | 75.9 | 79.3 |
| BERT-$A^3$ | 81.8 | 89.3 | 75.9 | 79.3 |
| BERT-JAT | 82.1 | 89.3 | 76.4 | 82.0 |
| $RoBERTa_{base}$ | 88.9 | 94.6 | 86.5 | 89.4 |
| RoBERTa-$A^3$ | 89.2 | 94.8 | 86.6 | 89.7 |
| RoBERTa-JAT | **90.1** | **95.2** | **87.0** | **90.2** |

parameters, which shows that JAT can capture higher-order dependencies than canonical self-attention. More specifically, BERT-JAT achieve 9.4% score rising on the CoLA dataset and 7.1% on the WNLI dataset as a remarkable improvement, which shows that JAT shows the better alignment ability on a smaller dataset. When it applies to RoBERTa, the average gain is larger and the most significant rising 4.9% happens on the CoLA dataset, which demonstrates a stable performance gain on SOTA models.

We have also show that JAT beats the competitor $A^3$ [24] generally. The $A^3$ introduces the triplet attention measuring *dissimilarity* to diversify the self-attention, while our proposed JAT tried to enhance *similarity* by building connections between unrelated pairs from a different perspective. These two works are orthogonal and can be performed together to improve self-attention mechanism. Meanwhile, the efficient variant JAT† also shows competitive performance than competitor $A^3$.

Results of evaluating JAT on the more challenging SuperGLUE benchmark are presented in Table 2, we have selected RoBERTa as a more efficient basic model. With more extended tokens on complicated tasks, the JAT fails on three tasks for OOM, but it still acquires better results than vanilla RoBERTa on others. The efficient variant RoBERTa-JAT† achieves the best average scores.

The result of the SQuAD experiment is summarized in Table 3. Our model BERT-JAT achieves better performance than all baselines in both EM and F1 scores, which is aligned with the above results.

## 4.3 Ablation Study

### 4.3.1 Effect of JAT's Layer Deployment

In this experiment, we explore the JAT layer's effect on acquiring the high-order connections. We set the layers at a different position in $RoBERTa_{base}$ model to use JAT heads, where the RoBERTa-$JAT_{l[i,j]}$ represents that the layers from $i$ to $j$ in $RoBERTa_{base}$ model have four JAT heads. We report the scores on the selected GLUE tasks, including CoLA, MRPC, RTE, and STS-B in Table 4. We can conclude that using JAT heads in lower or middle layers leads to better scores. This is consistent with the findings in Fig.(2), middle layers possess more weak high-order connections to be easily enhanced. Applying JAT at lower layers can effectively enhance the diversity of lower layers and improve the high-order connections to force the desired triadic closure.

### 4.3.2 Effect of JAT's Head Grouping

To evaluate the effect of JAT's head grouping (different splits) on general language understanding task accuracy, we trained many RoBERTa-JAT models with different JAT head numbers in layer 1-6. Results on the selected GLUE tasks are shown in Table 5. With the number of JAT head increasing, we can find that almost all the tasks' scores first increase then decrease. The scores increase because adding JAT heads enhance the high-order global dependencies in the vanilla RoBERTa model. They decrease when the number of heads is greater than 6, which probably is caused by too strong high-order inductive bias suppressing the canonical self-attention. But the CoLA dataset seems fitting the JAT assumption very well, which may explain why it reaches the highest performance gain in Table 1.

### 4.3.3 Effect of Higher Order in JAT

We perform this experiment to evaluate the effects of higher-order variants. Following the similar settings as the previous ablation study, where the RoBERTa-$JAT^{oi}$ stands for the $i$-th order self-attention defined in Eq.(9). We show the selected GLUE results in Table 6. And it reveals that for most natural language understanding tasks, the second-order self-attention is sufficient to discover

Table 4: The scores of different JAT's layer deployment.

| Model | CoLA | MRPC | RTE | STS-B |
|---|---|---|---|---|
| RoBERTa$_{base}$ | 63.6 | 90.2 | 78.7 | 91.2 |
| RoBERTa-JAT$_{l[1,4]}$ | 63.9 | **90.9** | 78.8 | **91.2** |
| RoBERTa-JAT$_{l[5,8]}$ | 63.6 | 90.4 | 78.7 | 90.9 |
| RoBERTa-JAT$_{l[9,12]}$ | 63.6 | 90.2 | 76.1 | 90.9 |
| RoBERTa-JAT$_{l[1,6]}$ | **64.6** | 90.4 | **79.4** | 91.1 |
| RoBERTa-JAT$_{l[7,12]}$ | 63.8 | 89.7 | 78.4 | 91.0 |

Table 5: The scores of different JAT's heads grouping.

| Model | CoLA | MRPC | RTE | STS-B |
|---|---|---|---|---|
| RoBERTa$_{base}$ | 63.6 | 90.2 | 78.7 | 91.2 |
| RoBERTa-JAT$_{h2}$ | 63.7 | **91.4** | 80.2 | **91.1** |
| RoBERTa-JAT$_{h4}$ | 63.9 | 91.2 | 80.2 | 90.6 |
| RoBERTa-JAT$_{h6}$ | 63.6 | 90.4 | 79.7 | 90.8 |
| RoBERTa-JAT$_{h8}$ | 63.8 | 90.2 | 78.9 | 90.7 |
| RoBERTa-JAT$_{h10}$ | 64.1 | 90.2 | 77.2 | 90.5 |
| RoBERTa-JAT$_{h12}$ | **64.7** | 90.0 | 77.8 | 90.5 |

Table 6: The scores of JAT's increasing order.

| Model | CoLA | MRPC | RTE | STS-B |
|---|---|---|---|---|
| RoBERTa$_{base}$ | 63.6 | 90.2 | 78.7 | 91.2 |
| RoBERTa-JAT$^{o1}$ | **65.4** | **91.4** | 79.9 | **91.3** |
| RoBERTa-JAT$^{o2}$ | 64.4 | 90.7 | **80.2** | 90.8 |
| RoBERTa-JAT$^{o3}$ | 64.0 | 91.2 | 78.7 | 90.9 |

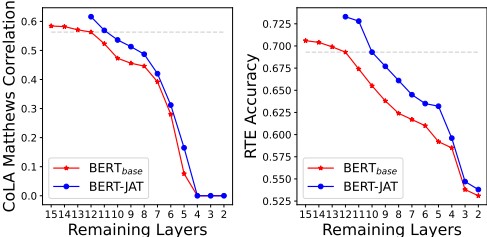

Figure 4: The performance decreases when the overall layers degrade from 12 to 2.

more higher-order connections and improve the expression ability of the model, which leads to better results. For example, using the second order JAT on RTE dataset significantly improves the scores.

#### 4.3.4 Effect of Layer Stacking Degradation in JAT

Similar to the evaluation in $A^3$ [24], we conduct layer stacking degradation experiments on both CoLA and RTE downstream tasks. Firstly, we reduce the simpler BERT$_{base}$ model by one layer at a time, which makes the overall layers degrade from 12 to 2. Fig.(4) shows that the BERT-JAT consistently outperforms BERT$_{base}$. In the RTE task, the BERT-JAT even achieves the highest accuracy with eight layers. Secondly, we have extended the BERT$_{base}$ model from 12 layers to 15 layers, which matches the comparable extra computation brought by the JAT mechanism (refer to Sec 4.6). It reveals that the deeper baseline lags behind the JAT-enhanced ones.

### 4.4 Visualization

We conduct a visualization in Fig.(5) to show the self-attention feature map changing in JAT. To avoid repetition, we construct a new sentence. The line between words represents their attention scores: the darker, the larger. The Fig.(5a) shows heads of layer 6 in BERT$_{base}$, and it indicates that the "Kate" has stronger connections to the words in the second sentence. The Fig.(5b) shows heads of the same layer in BERT-JAT, the "Kate" attends to the first sentence's "hit" through a "jump" connection with the blue head. This example shows that 'Kate' never gets a strong connection with the action 'hit' in vanilla self-attention, but JAT provides this conditional connection (may through 'bite'), which indicates the two actions may be sequentially related. This JAT-inspired attention change also applies to the 'SEP' token, which implies that 'Kate' are the last few words. Although this result differs from the previous examples on the noun exchanging, it reveals that the Jump Self-attention can enhance connections with various lexical patterns.

### 4.5 Case Study

We conduct an experiment on Named-Entity Recognition (NER) task using the standard CoNLL-2003 dataset, which concentrates on four types of named entities: persons, locations, organizations and names of miscellaneous entities that do not belong to the previous three groups. The settings are included in Appendix E. We select the major connections in the attention feature map where their scores are greater than (mean + std). We use 'CROSS' to represent connections of tokens with different named-entities. The visualization is showed in Fig.(7). From Fig.(7a) we find there are not many 'CROSS' connections in the attention of BERT$_{base}$, and most of the connections are between tokens belong to person entity. From Fig.(7b), we observe JAT significantly increases the number of connections between tokens with different named-entities at the layers {1,2,3,8,9,10} having JAT heads, which enhances the model's ability to discover high-order information and makes the model's expression more diverse.

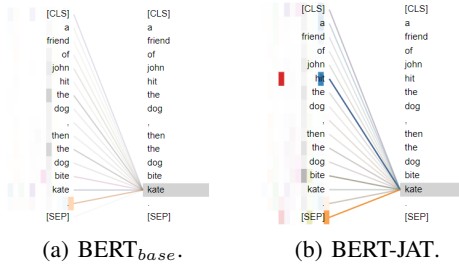

(a) BERT_base.      (b) BERT-JAT.

Figure 5: The sentence is "A friend of John hit the dog, then the dog bite Kate.", our BERT-JAT finds the "jump" connection between word "hit" and word "Kate" in 6-th Layer.

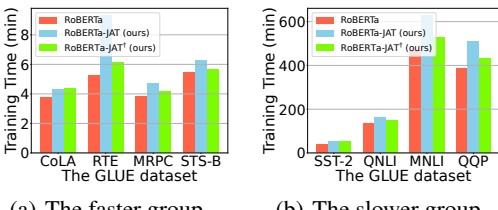

(a) The faster group.    (b) The slower group.

Figure 6: The run-time of the proposed Jump Self-attention (JAT, JAT$^{\dagger}$) on the RoBERTa model. We divided eight tasks into a faster ($< 100$ min) and a slower group for clarity. Each bar stands for the training process on individual downstream tasks (running 10 epochs).

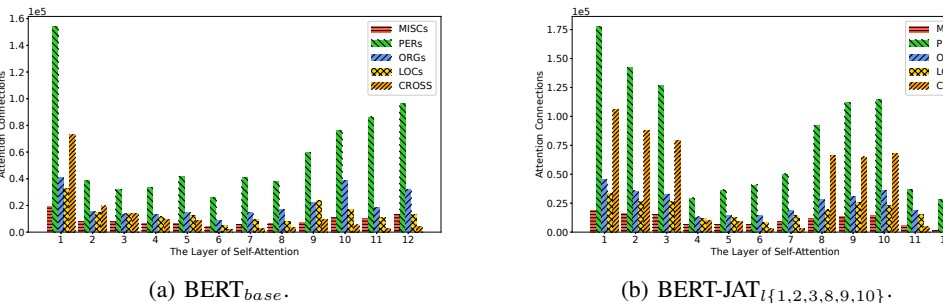

(a) BERT_base.            (b) BERT-JAT$_{l\{1,2,3,8,9,10\}}$.

Figure 7: The pair-wise attention connections of BERT$_{base}$ and BERT-JAT on dataset CoNLL2003, our model BERT-JAT improve the connection between tokens with different named-entities (the 'CROSS' connections in orange).

## 4.6 Run-time Comparison

With the same setting, we evaluate the iterative efficiency of JAT on eight downstream task, which has no learnable parameters. The Fig.(6) shows that JAT mostly brings about 30% extra computation than the vanilla Transformer, where RTE achieves 75% at most. This variance is caused by the different input lengths $L$ when calculating the adjacency matrix in Eq.(3). It could be further improved by utilizing the Keys' sparsity [25], then JAT$^{\dagger}$ achieves no more than 15% extra computation time.

## 5 Conclusions

In this paper, we discovered the existence of Jump Self-attention (JAT) in the Transformers' dot-product computation. Drawing inspirations from the pieces moving of English Draughts, we define JAT from the canonical self-attention and propose a tangible framework to formalize JAT into the graph learning problem and leverage it to encourage JAT's propagation in attention feature map. We further develop the higher-order variants under multi-hop assumption to increase the generality. The proposed architecture is compatible with the pre-trained models. Experiments on two tasks demonstrate JAT's superior performance on capturing high-order dependencies. We also want to use JAT on MindSpore (`https://www.mindspore.cn/`), which is a new deep learning computing framework.

## Acknowledgments

This work was supported by grants from the Natural Science Foundation of China (62202029 and U20B2053). Thanks for the computing infrastructure provided by Beijing Advanced Innovation Center for Big Data and Brain Computing. This work was also sponsored by CAAI-Huawei MindSpore Open Fund. We gratefully acknowledge the support of MindSpore, CANN (Compute Architecture for Neural Networks) and Ascend AI Processor used for this research.

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
