# OpenReview forum: "Jump Self-attention: Capturing High-order Statistics in Transformers"
_NeurIPS.cc/2022/Conference — NeurIPS 2022 Accept_

### Official Review · Reviewer_YzKV · 2022-06-30

**Rating:** 7
**Confidence:** 4
**Soundness:** 3 good
**Presentation:** 2 fair
**Contribution:** 3 good

**Summary:**

This paper proposes Jump self-attention(JAT), which enhances first-order(i.e. dot-product of two hidden states) self-attention by running GCN
on its attention score map to capture high-order statistics. JAT can be interchangeably used with the canonical self-attention, and could be further generalized from second-order to higher orders. An efficient variant of JAT is proposed to avoid the extra computation cost. Experiments on GLUE and SQUAD show that JAT significantly increase the performance of BERT/RoBERTa-base with only 15% extra computation and no new parameters.

**Questions:**

## Notations
Definition 3.1 is confusing: are vector inputs x_i, x_j, x_k fixed vectors? If they are
fixed, what does `Pr[\dot]` mean? If they are random varaibles, what's the distributions?
The same question applies to weight matrix W_q and W_k.

Section 3.2 is very hard to follow. For example, what is `adjacency matrix A` in #L159?
If it's the same matrix defined in Eq.(3), it's better to introduce that you're going to compute its components
earlier instead of letting reader find it themselves after 10+ lines.
I think a more detailed explanation of brute-force
method in #L151~#L153 could help readers get better understanding on motivations
and the derivation of matrix A in Eq.(3)

There are two `BERT-JAT(ours)` and two `RoBERTa-JAT(ours)` in Table1, what's the difference between them?

## Experiments
An assumption of mine is that two layers of one-order attention
could simulate a single layer two-order attention. Therefore I'd like
to see more experiments where authors use BERT/RoBERTa-large and
test whether the improvements of jump-attention still exist.


**Limitations:**



**Strengths And Weaknesses:**

### Strengths
* This paper proposes novel and simple module to enhance performance of canonical self-attention with no new learnable parameters.
* Experimental results show significant improvements over BERT/RoBERTa baselines, and authors provide ablation study on how to
tune the hyperparams.


### Weaknesses
* Writing need improvement. The method part is quite hard to follow.
* There is no experiments on BERT/RoBERTa-large models.


I'd like to increase my score if the two weaknesses above are addressed.

---

> ### Author Response · Authors · 2022-08-02
> **Response to Reviewer YzKV**
>
> We would like to thank all the reviewers for the constructive feedback, which we will leverage to improve this work in the rebuttal version and future version. We appreciate R4’s positive comments, such as the novelty, simplicity, and significant improvement. And we will address your concerns one by one.
>
> **Q1 Improve the method parts:**
>
> Thanks for your advice. We have reorganized the methods part in the rebuttal version. We also include a `symbol list table` in the rebuttal version (Appendix) to clarify the mathematical notions. And we draw a `network architecture figure` to better help the understanding of the whole process in the rebuttal version (Appendix).
>
> **Q2 Experiments on BERT/RoBERTa-large models:**
>
> Thanks for the kind suggestions. Due to the hardware limitation, we can only present incomplete results on large models. We are trying to get help from facilities like Hugginface, and include more results in the final version. Here is the peek table. The experiment is carried out on the SOTA model, RoBERTa-Large. We have witnessed a constant performance improvement on large models, which indicates that the JAT could break the dot-product limitation of canonical self-attention.
>
> |  | CoLA | MRPC | RTE | STS-B | SST-2 |
> | --- | --- | --- | --- | --- | --- |
> | RoBERTa_base | 63.6 | 90.2 | 78.7 | 91.2 | 94.8 |
> | RoBERTa_Large | 68.0 | 91.4 | 86.8 | 92.4 | 96.4 |
> | + JAT | 69.1 | 90.2 | Out of Memory | Out of Memory | 96 |
> | + JAT$^{+}$ | **70.2*8 | **92.2** | **88.1** | **92.8** | **96.8** |
>
> **Q3 The details of Def 3.1:**
>
> Sorry for causing this misunderstanding. The vector inputs $\mathbf{x}_i$, $\mathbf{x}_j$, $\mathbf{x}_k$ are indeed vectors, but we could consider them as random variables whose distributions are mathematically intractable. The transformation matrix $\mathrm{\mathbf{W}}_Q$ and $\mathrm{\mathbf{W}}_K$ are generated from another distribution. We have the combination of random variables, the attention score $\mathrm{\mathbf{S}} {}_i {}_j=(\mathbf{x}_i \mathrm{\mathbf{W}}_Q)(\mathbf{x}_j \mathrm{\mathbf{W}}_K)^{\top}$, and we are investigating the probability of the event $S {}_i {}_j \geq p$. This is the probabilistic interpretation of self-attention calculation, which is also discussed as deterministic mapping in [1].
>
> [1] An, Bang, et al. "Repulsive Attention: Rethinking Multi-head Attention as Bayesian Inference." *EMNLP* 2020.
>
>
> **Q4 Reorganize the Section 3.2:**
>
> Thanks for your advice. First, the adjacency matrix A is computed on an individual key and is merged through a summation, we have adjusted the corresponding description. Actually, the brute-force method is the way we performed in the case study in Figure 2. We have reorganized that part.
>
> **Q5 Difference between two JAT methods:**
>
> Sorry for causing such confusion. The performance of JAT$^{+}$ has been underestimated in the previous submission. After we have corrected the code implementation, it shows totally comparable performance, e.g., avg 87.1=87.1. It uses the sparsity measurement from the Informer work, and helps remove redundant Keys before the keys summation in Eq.(3). A less noise adjacency matrix may contribute well to capturing high-order statistics. We have updated Table 1 in the rebuttal version. Here is the peek table.
>
> |  | CoLA | MRPC | RTE | STS-B | QNLI | QQP | SST-2 | MNLI | Average |
> | --- | --- | --- | --- | --- | --- | --- | --- | --- | --- |
> | RoBERTa | 63.6 | 90.2 | 78.7 | 91.2 | 92.8 | **91.9** | 94.8 | 87.6 | 86.4 |
> | + JAT | 65.9 | 91.4 | **80.2** | 91.3 | **93.2** | 91.7 | **95.4** | **87.7** | **87.1** |
> | + JAT$^{+}$ | **66.7** | **91.7** | 79.2 | **92.3** | 92.9 | 91.7 | 94.9 | 87.5 | **87.1** |

---

> > ### Comment · Reviewer_YzKV · 2022-08-08
> > **Feedback**
> >
> > Thank you for your thorough responses to my feedback. I increased my score to7. Looking forward to a more easy-to-follow version of your paper and your latest results on large models!

---

> > > ### Author Response · Authors · 2022-08-08
> > > **Response to Feedback**
> > >
> > > Dear reviewer YzKV,
> > >
> > > Thanks for your kind feedback, and we will try to build the final version in a more easy-to-follow manner, together with complete results on large models.
> > >
> > > Please feel free to let us know if you have any questions or further suggestions.
> > >
> > > Yours,
> > >
> > > Authors

---

### Official Review · Reviewer_mSLD · 2022-06-30

**Rating:** 6
**Confidence:** 4
**Soundness:** 3 good
**Presentation:** 3 good
**Contribution:** 3 good

**Summary:**

In this work, the authors found that the high-order self-attention did exist in the dot-product computation but it was too weak to contribute much to the attention layer’s output. Take two sentences “Book costs more than pencil.” and “Book costs less than computer.” as an example. “Computer” never attends to “pencil” until they were both compared with “Book”, thus “Book” is the “stepping stones” in creating high-order attention between “computer” and “pencil”. In order to capture such high-order dependency in Transformer model, this paper proposed Jump Self-attention (JAT), and the main contribution can be summarized as:

1. The authors proposed the JAT through the jump aggregation and high-order dependencies’ propagation without additional network parameters, and the efficient variant avoided the extra computation cost.
2. The authors proposed the higher-order variants of JAT and the masking strategy to allow its interchangeable usage with the common self-attention on the pre-trained models.
3. The experiments were conducted on ten different tasks to examine the effectiveness of the proposed methods. Results demonstrated the success of JAT in achieving state-of-the-art performance.

**Questions:**

The paper is clearly written. But, the game the authors mentioned was not popular.

**Ethics Review Area:**

["I don’t know"]

**Limitations:**

The improvement is limited when compared with the related works.

**Strengths And Weaknesses:**

Pros:
1. The motivation of capturing high-order dependency was reasonable and very intuitive and the example they provided could help clarify concepts.
2. This paper proposed the idea of capturing high-order dependency by measuring the similarity between different queries and keys, and then used the GCN to enhance this kind of connection. The idea is novel, and the result showed that the mothed could outperform canonical self-attention.
3. This study designed several ablation studies to explan how their method worked. The run-time comparison was used to show the effect of efficient variants.

Cons:
1. They said that they were inspired by the pieces moving of English Draughts, although they spent some space explaining the rules, However, this game is not so well-known, and may still cause some confusion.
2. Although the efficient variants could indeed reduce the computation cost but still brought extra computation relative to the vanilla Transformer. And, the improvement from efficient variants to original self-attention was limited.
3. This paper mentioned “miracle heads” several times, but the authors didn’t do comparison among these related methods. The authors may show the performance and computation cost of finding miracle heads.

---

> ### Author Response · Authors · 2022-08-02
> **Response to Reviewer mSLD**
>
> We would like to thank all the reviewers for the constructive feedback, which we will leverage to improve this work in the rebuttal version and future version. We appreciate R3’s positive comments, such as the reasonable motivation, intuitive example, and novelty. And we will address your concerns one by one.
>
> **Q1 The piece moving example:**
>
> Sorry for causing such confusion. We recall the English Draughts, also checkers (many variants worldwide), to show the core idea of proposed methods better — `Jump` attention. The canonical self-attention only builds direct connections through pairwise similarity, and the proposed JAT attention tries to acquire the “jump” connections by playing like the piece moving. We hope this game example could help the readers get closer to the main contribution.
>
> **Q2 The performance of efficient variants:**
>
> Sorry for causing such confusion. The performance of JAT$^{+}$ has been underestimated in the previous submission. After we have corrected the code implementation, it shows totally comparable performance, e.g., avg 87.1=87.1. It uses the sparsity measurement from the Informer work, and helps remove redundant Keys before the keys summation in Eq.(3). A less noise adjacency matrix may contribute well to capturing high-order statistics. We have updated Table 1 in the rebuttal version. Here is the peek table.
>
> |  | CoLA | MRPC | RTE | STS-B | QNLI | QQP | SST-2 | MNLI | Average |
> | --- | --- | --- | --- | --- | --- | --- | --- | --- | --- |
> | RoBERTa | 63.6 | 90.2 | 78.7 | 91.2 | 92.8 | **91.9** | 94.8 | 87.6 | 86.4 |
> | + JAT | 65.9 | 91.4 | **80.2** | 91.3 | **93.2** | 91.7 | **95.4** | **87.7** | **87.1** |
> | + JAT$^{+}$ | **66.7** | **91.7** | 79.2 | **92.3** | 92.9 | 91.7 | 94.9 | 87.5 | **87.1** |
>
> **Q3 The miracle heads:**
>
> Sorry for causing such a misunderstanding. The miracle heads refer to these most effective heads in capturing the jump self-attention, which is hardly acquired in the standard Transformer architecture. If the magnitude of jump connections $\rho$ is fixed, finding the statistics are combination problem. But the optimization becomes intractable. Meanwhile, we are the first to define jump attention. Thus, the comparison should be natural. We have clarified that concept in the rebuttal version.
>
> **Q4 Game is not popular:**
>
> Please refer to Q1.
>
> **Q5 Limited performance:**
>
> Sorry for causing such an impression. We would like to point out that the improvement is significant. First, we evaluate the proposed JAT on the SOTA model, RoBERTa. Then we have improved all the down streaming tasks with the same modifications, and constant beats the related works. Third, the performance could be further improved by training from scratch.
>
> Due to the hardware limitation and lack of data, we follow the commonly pre-trained and fine-tuning framework. To be more specific, we load the architecture and weights from the pre-trained model, then we replace the corresponding heads with the JAT one. And the final results are acquired from the fine-tune models. We suspect the improvement problem to the fine-tune framework, and we have included one `training from the scratch` result in the first submission, which reveals the potential of building models with JAT on the CoLA dataset. Here is the peek table. Firstly, we can see that the RoBERTa model receives constant gains after applying the JAT. Secondly, the training from the scratch (`tfs`) result achieves better performance than our best reporting fine-tuning (`ft`) one, namely 66.5 > 65.4.
>
> | CoLA data % | 20% | 40% | 60% | 80% | 100% |
> | --- | --- | --- | --- | --- | --- |
> | RoBERTa (ft) | - | - | - | - | 63.6 |
> | RoBERTa-JAT (ft) | - | - | - | - | 65.4 |
> | RoBERTa (tfs) | 51.0 | 55.0 | 59.4 | 60.1 | 63.7 |
> | RoBERTa-JAT (tfs) | 52.5 | 56.7 | 60.6 | 60.9 | **66.5** |
> | $\Delta$ Performance | +1.5 | +1.7 | +1.2 | +0.8 | +0.8 |
>
> We are trying to contact some organizations like Huggingface to obtain more large-scale computing resource, and to include more promising results in the final version.

---

### Official Review · Reviewer_5zAj · 2022-07-11

**Rating:** 6
**Confidence:** 4
**Soundness:** 3 good
**Presentation:** 3 good
**Contribution:** 3 good

**Summary:**

This paper proposes a method to model jump attention in sequence data (e.g., in the sentence “Jack is the father of John and John is the brother of Mary”, we have a jump connection between Jack and Mary using the stepping stone of John). Specifically, the authors add a one-layer GCN (without activation function) after a self-attention layer. The construction of the graph is based on the attention score (QK). In multi-head attention, parts of the heads are applied to the jump attention while the remaining preserve, making the network able to capture both first- and higher-order connection. To reduce parameters, the parameters of the GCN is merged into parameters of the self-attention layer. To reduce computation cost, the authors apply an existing sparsity measurement. The method is validated on GLUE and SQuAD comparing with baseline models and a recently proposed A3 method.

This is an interesting work with clear and convincing motivation, and it also reads smoothly. It is true that entities in texts may have relations without direct connection. The pivot entities (or say, stepping stones) become important under this case. Though GCN itself is not new, the definition of high-order connection (Definition 3.1 and Section 3.2.3) and the parameter-merging step in Eq (5) is novel. The authors find that the two parameter matrices of GCN and Attention can be merged (because they are both linear transformation).

**Questions:**

I have some questions for you. Correct me if I’m wrong:
1.	The definition of the adjacency matrix A. From Eq (2), we can see if S_{ij} * S_{kj} is greater than the threshold \rho, then i and k are considered as connected nodes. But this definition is a little bit different from Definition 3.1 saying both S_{ij} and S_{kj} need to be greater than \rho. Can you explain more on this point?
2.	In the combination of GCN and Self-Attention, the equations seem to omit the activation function of the GCN. Is it merged to the activation function of the later Feed-Forward Net? Or the ignoration makes no significant difference? It is better to elaborate the point clear in the method.
3.	What’s the effect of tuning the hyper-parameter \rho?
4.	The explanation of Fig. 4. I think it will be better if the definition of connection can be linked to some objectives or downstream tasks because it is hard to tell directly how “hit” is related to “Kate”. Also, in this example, it is unclear when we need jump connection or just first-order connection. For example in “The dog bite Kate”, is the “dog” directly related to “Kate” or they need a stepping stone “bite”?
5.	In Line 272, “almost all the tasks’ scores first increase then decrease” may not hold since in Table 4, the main trend is just decreasing. I think it is better to add a RoBERTa model without JAT for comparison.
6.	How slower it will be if not applying the sparsity measurement is shown in Fig. 5, but I wonder how much the performance will drop/increase.


**Limitations:**

Minor suggestions:
1.	The writing can be further improved. There are typos in:
a)	Line 121, “significant \delta” should be “significance \delta”.
b)	Line 162, “is a symmetric matrix contains input” should be “is a symmetric matrix containing input”.
c)	Line 176-177, “for encourage the attended items becomes” should be “for encouraging the attended items to become”.
2.	The captions of Fig. 1 and Fig. 2 have large overlaps with your content. You can consider shrinking the captions to leave more space to your methods or related work.

**Strengths And Weaknesses:**

Strengths:

+ Good readability. Same case study across the paper.
+ Convincing motivation of jump connection.
+ Novel method to combine GCN and Self-Attention.

Weaknesses:

- Lack information of comparison with related work.
- Experiments on SQuAD have no results of the competitor A3.
- Improvement is limited.


UPDATE:
The authors' responses address my main concerns, which should be included in the revised version.

---

> ### Author Response · Authors · 2022-08-02
> **Response to Reviewer 5zAj (Part 1)**
>
> We would like to thank all the reviewers for the constructive feedback, which we will leverage to improve this work in the rebuttal version and future version. We appreciate R2’s positive comments, such as the convincing motivation, good readability, and novelty. And we will address your concerns one by one.
>
> **Q1 Lack information of comparison with related work:**
>
> Due to space limitation, we have included the related work section in the Appendix (supplementary material). And we added a more detailed discussion on the related works in #Lines 501-505. We will move the discussion of related work into the main paper from the Appendix in the final version.
>
> **Q2 Experiments on SQuAD have no results of the competitor A3:**
>
> Thanks for your corrections. We have added the competitor results in the rebuttal version. Here is the peek table. Our proposed JAT model shows superior performance than the most related competitor A^3, on the QA tasks. We deduced this to the JAT’s proper definition of high-order statistics rather than the coarse diversity in A^3.
>
> | Methods | SQuAD v1.1 (EM) | SQuAD v1.1 (F1) | SQuAD v2.0 (EM) | SQuAD v2.0 (F1) |
> | --- | --- | --- | --- | --- |
> | BERT_base | 81.2 | 87.9 | 75.9 | 79.3 |
> | BERT-A3 | 81.8 | 89.3 | 74.1 | 78.3 |
> | BERT-JAT (ours) | 82.1 | 89.3 | 76.4 | 82.0 |
> | RoBERTa_base | 88.9 | 94.6 | 86.5 | 89.4 |
> | RoBERTa-A3 | 89.2 | 94.8 | 86.6 | 89.7 |
> | RoBERTa-JAT (ours) | **90.1** | **95.2** | **87.0** | **90.2** |
>
> **Q3 Improvement is limited:**
>
> Sorry for causing such an impression. We would like to point out that the improvement is significant. First, we evaluate the proposed JAT on the SOTA model, RoBERTa. Then we have improved all the down streaming tasks with the same modifications, and constant beats the related works. Third, the performance could be further improved by training from scratch.
>
> Due to the hardware limitation and lack of data, we follow the commonly pre-trained and fine-tuning framework. To be more specific, we load the architecture and weights from the pre-trained model, then we replace the corresponding heads with the JAT one. And the final results are acquired from the fine-tune models. We suspect the improvement problem to the fine-tune framework, and we have included one `training from the scratch` result in the first submission, which reveals the potential of building models with JAT on the CoLA dataset. Here is the peek table. Firstly, we can see that the RoBERTa model receives constant gains after applying the JAT. Secondly, the training from the scratch (`tfs`) result achieves better performance than our best reporting fine-tuning (`ft`) one, namely 66.5 > 65.4.
>
> | CoLA data % | 20% | 40% | 60% | 80% | 100% |
> | --- | --- | --- | --- | --- | --- |
> | RoBERTa (ft) | - | - | - | - | 63.6 |
> | RoBERTa-JAT (ft) | - | - | - | - | 65.4 |
> | RoBERTa (tfs) | 51.0 | 55.0 | 59.4 | 60.1 | 63.7 |
> | RoBERTa-JAT (tfs) | 52.5 | 56.7 | 60.6 | 60.9 | **66.5** |
> | $\Delta$ Performance | +1.5 | +1.7 | +1.2 | +0.8 | +0.8 |
>
> We are trying to contact some organizations like Huggingface to obtain more large-scale computing resource, and to include more promising results in the final version.
>
> **Q4 Explanation of the difference between Eq.(2) and Def 3.1:**
>
> Sorry for causing such a misunderstanding. The derivation of Eq.(2) has a small gap from Def 3.1, where the symbol is different, $p$ in Def 3.1 while $\rho$ in Eq.(2). The $\rho$ is a soft controller than $p$. As we have stated in #Lines 148-151, if we go through all pairs of Q and K, it may have exact high order similarity as it is in Def 3.1. But the direct modification as previously defined makes it a combination problem, which also requires $O(L^3)$ computation. We slightly break `both` requirements on $S_{ij}$ and $S_{kj}$, and relax them to encourage their product to be large.  Then we leverage the GCN trick to enhance the weak jump connections rather than direct modification. Thus, it becomes a network component with BP-friendly characteristics. On the other hand, the small gap may allow more flexible intermediate tokens than the strict definition when the “side effects” help in our empirical evaluation.  Ignoring the small gap helps Eq.(2) remain simple. We have rephrased this part in the rebuttal version. Note that we also evaluated by dropping out the negative case.

---

> ### Author Response · Authors · 2022-08-02
> **Response to Reviewer 5zAj (Part 2)**
>
> (continue with the previous one)
>
> **Q5 The omitting of the activation function:**
>
> Thanks for the kind suggestions. We only use the GCN’s aggregation ability to manipulate the self-attention scores. As the reviewer mentioned, the activation function could be merged to the later Feed-Forward Net. Besides, the first step of following the self-attention operation is the row-wise Softmax(). If we add the activation function, it will hurts the normalization effects. We have rephrased this part in the rebuttal version.
>
> **Q6 The effect of tuning the hyper-parameter \rho:**
>
> We will update the Appendix with a parameter-sensitive section. The hyperparameter $\rho$ decides the magnitude of building jump attention, which may vary across the dataset. From the empirical evaluation, it generalizes well from 2 to 6, and a parameter search is needed in practical usage. Here is the peek table. It can be seen that the performance is relatively stable w.r.t $\rho$ changing.
>
> `RoBERTa_base + JAT`
>
> | $\rho=$ | 2 | 2.5 | 3 | 3.5 | 4 | 4.5 | 5 | 5.5 | 6 |
> | --- | --- | --- | --- | --- | --- | --- | --- | --- | --- |
> | CoLA | 0.640 | 0.641 | 0.633 | 0.644 | 0.641 | 0.636 | **0.659** | 0.633 | 0.638 |
> | MRPC | 0.902 | 0.904 | 0.902 | 0.909 | **0.914** | 0.902 | 0.907 | 0.902 | 0.895 |
> | RTE | 0.773  | 0.791  | 0.780  | 0.773  | 0.783  | 0.765  | 0.780  | **0.792**  | 0.787  |
> | STS-B | **0.913** | 0.911 | 0.911 | 0.911 | 0.910 | 0.910 | 0.912 | 0.911 | 0.909 |
>
> `RoBERTa_base + JAT$^{+}$`
>
> | $\rho=$ | 2 | 2.5 | 3 | 3.5 | 4 | 4.5 | 5 | 5.5 | 6 |
> | --- | --- | --- | --- | --- | --- | --- | --- | --- | --- |
> | CoLA | 0.646 | 0.655 | **0.667**  | 0.639 | 0.659  | 0.648 | 0.654  | 0.635 | 0.654 |
> | MRPC | 0.914 | **0.917** | 0.900 | 0.907 | 0.909 | 0.900 | 0.907 | 0.900 | 0.912 |
> | RTE | 0.794 | 0.791 | **0.809** | 0.794 | 0.791 | 0.780 | 0.787 | 0.809 | 0.798 |
> | STS-B | 0.911 | 0.912 | 0.911 | 0.912 | 0.911 | 0.911 | 0.912 | **0.923** | 0.911 |
>
> **Q7 The explanation of Fig. 4:**
>
> Thanks for the kind suggestions. The model is trained on the QA dataset, and we only used the encoder part to show the connection between `dog` and `kate`. Since the downstream tasks are not like the NER case study, the JAT model may focus on different stepping stone words. We will clarify this in the final version.
>
> **Q8 Add basic RoBERTa to Table 4:**
>
> Thanks for your advice. We have put the RoBERTa’s results in the rebuttal version, and rephrased the corresponding discussion for better comparison clarity.
>
> **Q9 The performance change after applying the sparsity measurement:**
>
> Sorry for causing such confusion. The performance of JAT$^{+}$ has been underestimated in the previous submission. After we have corrected the code implementation, it shows totally comparable performance, e.g., avg 87.1=87.1. It uses the sparsity measurement from the Informer work, and helps remove redundant Keys before the keys summation in Eq.(3). A less noise adjacency matrix may contribute well to capturing high-order statistics. We have updated Table 1 in the rebuttal version. Here is the peek table.
>
> |  | CoLA | MRPC | RTE | STS-B | QNLI | QQP | SST-2 | MNLI | Average |
> | --- | --- | --- | --- | --- | --- | --- | --- | --- | --- |
> | RoBERTa | 63.6 | 90.2 | 78.7 | 91.2 | 92.8 | **91.9** | 94.8 | 87.6 | 86.4 |
> | + JAT | 65.9 | 91.4 | **80.2** | 91.3 | **93.2** | 91.7 | **95.4** | **87.7** | **87.1** |
> | + JAT$^{+}$ | **66.7** | **91.7** | 79.2 | **92.3** | 92.9 | 91.7 | 94.9 | 87.5 | **87.1** |
>
> **Q10 Minors:**
>
> Thanks for the corrections. We have accepted and fixed them in the rebuttal version.

---

> ### Author Response · Authors · 2022-08-08
> **[Message to Reviewer 5zAj] Reminder for Author-Reviewer discussion**
>
> Dear reviewer 5zAj],
>
> Thank you very much for the constructive suggestions, I really enjoy reading and learning from your comments.
>
> Since the deadline for author-reviewer is one day away. May I know if our response has addressed your questions?
> Please feel free to let us know if you have any questions. We are very much looking forward to having the opportunity to discuss those with you.
>
> Best,
>
> Authors

---

> > ### Comment · Reviewer_5zAj · 2022-08-09
> > **Thanks for the detailed response.**
> >
> > I will increase my score to 6.

---

> > > ### Author Response · Authors · 2022-08-09
> > > **Thank you! It was the least I could do.**
> > >
> > > Thanks for your kind feedback.
> > > Please feel free to let us know if you have any questions or further suggestions.
> > >
> > > Yours,
> > >
> > > Authors

---

### Official Review · Reviewer_MTPd · 2022-07-16

**Rating:** 6
**Confidence:** 4
**Soundness:** 3 good
**Presentation:** 3 good
**Contribution:** 3 good

**Summary:**

The paper proposes a new self-attention mechanism that generalizes attention to a graph and takes into account connections with an intermediate node. The authors develop the intuition and frame their work in the context of prior work. They then perform experiments that show that their attention mechanism does indeed help the model perform better on more challenging linguistic tasks.

**Questions:**

They mention that their method increases computation by 30% or 15% depending on the configuration. That seems equivalent to adding an additional 1-3 layers to BERT. Does their method still perform well against a bigger BERT?

Since their attention mechanism captures higher-order relationships, does it do well with on question-answering tasks that require multi-hop reasoning?

What is the relative RAM requirements of their mechanism? How does it scale with L? The authors only address computation.

**Limitations:**

Possibly, their method becomes harder to apply on longer sequences.

**Strengths And Weaknesses:**

Originality: The idea has roots in triplet attention and Graph Convolutional Networks which the authors mention. It does appear to be an original application to my knowledge.

Clarity: I like how the authors develop the intuition with the computer-pencil example. I was a little confused for some time because I expected to see a direct edge, but I think the authors meant they are connected only through the intermediate book token? The formal definitions and algorithms are clearly defined and motivated;

Quality: Experiments are well done. I think there could be more tasks and performance benchmarks, though.

Significance: It's interesting to see that there does indeed appear to be better performance in the more challenging linguistic tasks, so I think the authors' intuitions seem to bear out and the method is efficient enough to be generally usable.

---

> ### Author Response · Authors · 2022-08-02
> **Response to Reviewer MTPd (Part 1)**
>
> We would like to thank all the reviewers for the constructive feedback, which we will leverage to improve this work in the rebuttal version and future version. We appreciate R1’s positive comments, such as the original application, intuitive example, and efficient method. And we will address your concerns one by one.
>
> **Q1 Confused about the computer-pencil example:**
>
> Sorry for causing this misunderstanding. We consider this as a logical reasoning question about the price comparison (#Line 86). It is assumed that the `book` and `pencil` are compared in the first sentence, while only the `book` and `computer` are compared in the second sentence. The example focuses on “recognizing the valuable relationship between the three different objects”, and we want to find the connection between the `pencil` and `computer` through the intermediate `book` token. Note that this may not be the only selection of intermediate tokens, and we just try to simplify the problem first, where the proposed JAT allows any high-order similarity with fixed significant threshold $p$. We have reorganized the paragraphs to show better the example’s goal (#Lines 85-112). In section 4.5, we perform a case study to further investigate the Named-Entity Recognition task as an empirical evaluation of the intuitive example.
>
> **Q2 More tasks and benchmarks:**
>
> Thanks for the kind suggestions. We argue that extensive experiments have already been performed on the commonly used GLUE (Table 1) and SQuAD V1/V2 (Table 2) benchmarks. Although the proposed JAT is applied to the natural language understanding (NLU) tasks, we believe running all the NLU benchmarks is unnecessary to demonstrate JAT’s effectiveness. Besides, we only afford the latest DeBERTa model on the GLUE benchmarks with `very coarsely fine-tuned` due to the hardware and runtime limitations. Here are our current results. We are finding a way to verify the DeBERTa model on the superGLUE benchmarks, and we will update it during the rolling discussion.
>
> | GLUE | CoLA | MRPC | RTE | STS-B |
> | --- | --- | --- | --- | --- |
> | DeBERTa-base | 63.8 | 90.2 | 74.4 | 91.2 |
> | DeBERTa-JAT | 65.6 | 90.4 | 74.7 | 91.3 |
> | DeBERTa-JAT$^{+}$ | **65.8** | **90.7** | **78.7** | **91.5** |
>
> **Q3 Add 15%+ Deeper BERT as baselines:**
>
> Thanks for the kind suggestions to make our comparison complete. Before diving into the empirical results, we would like to explain some details of our experiments. Taking the BERT as an example, our experiments are based on the public pre-trained model ([https://huggingface.co/bert-base-uncased](https://huggingface.co/bert-base-uncased)). Some intermediate heads are replaced by the JAT ones with initial weights, and we performed a fine-tuning procedure on CoLA and RTE datasets as the final models. So we believe that adding 1-3 plain layers is fair experiment settings rather than extracting weights from the larger BERT pre-trained models. Here are our current results. From the comparison below, we can conclude that our proposed JAT always shows superior performance to the basic model. Meanwhile, even though the basic model is equipped with more layers, its performance still falls behind, e.g., 0.542.
>
> | CoLA | 2 | 3 | 4 | 5 | 6 | 7 | 8 | 9 | 10 | 11 | 12 | 13 | 14 | 15 |
> | --- | --- | --- | --- | --- | --- | --- | --- | --- | --- | --- | --- | --- | --- | --- |
> | BERT | 0.0 | 0.0 | 0.0 | 0.076  | 0.280  | 0.392  | 0.446  | 0.456  | 0.473  | 0.489  | 0.522  | 0.537  | 0.542  | 0.542 |
> | BERT-JAT | 0.0 | 0.0 | 0.0 | 0.165  | 0.312  | 0.420 | 0.487  | 0.513  | 0.516 | 0.519  | **0.552**  | - | - | - |
>
> | RTE | 2 | 3 | 4 | 5 | 6 | 7 | 8 | 9 | 10 | 11 | 12 | 13 | 14 | 15 |
> | --- | --- | --- | --- | --- | --- | --- | --- | --- | --- | --- | --- | --- | --- | --- |
> | BERT | 0.531  | 0.538  | 0.585  | 0.592  | 0.610  | 0.610  | 0.614  | 0.628  | 0.635  | 0.592  | 0.635  | 0.606  | 0.606  | 0.603  |
> | BERT-JAT | 0.538 | 0.547  | 0.596  | 0.632  | 0.635  | 0.625  | 0.661  | 0.639  | 0.639  | 0.621  | **0.628**  | - | - | - |
>
> **Q4 The performance on QA tasks:**
>
> Sorry for causing this misunderstanding. Due to the space limitation, we have presented the results on SQuAD v1.1/v2.0 in Table 2, which is aligned with other ablation study tables. We also include the corresponding description on #Lines 258-259. Compared with the BERT and RoBERTa model, our proposed JAT enhanced model has an average improvement of 0.92% (EM) and 1.63% (F1). The most significant improvement, namely 3.40% (F1), is achieved by evaluating the BERT mode on the SQuAD v2.0 dataset. Combined with the higher order (multi-hop variants) ablation study in section 4.3.3, the second-order hops are enough for most tasks. In summary, we believe JAT could help improve the multi-hop reasoning on question-answer tasks. We will try to include some QA examples in the final version (Appendix).

---

> ### Author Response · Authors · 2022-08-02
> **Response to Reviewer MTPd (Part 2)**
>
> (continue with the previous one)
>
> **Q5 The space overhead of JAT:**
>
> The efficient variant JAT$^{+}$ mainly reduced the computation complexity from $O(L^3)$ to $O(L^2)$, which is comparable with the canonical self-attention. Since we have merged the graph filter $\mathcal{W}$ into the projection matrix $\mathbf{W}$ in Eq.(5) at #Lines 184-186, the learning parameters will not change. The calculation of GCN is linear with $L$, but there exists extra space overhead, where its analysis can be divided into two parts. The first is the attention mechanism, which makes the same level $O(L^2)$ as it in the canonical formulation. We can leverage better speed-up methods if needed. And the second one is the adjacency matrix computing before performing the GCN trick, i.e., $O(L^2)$, which is considered pre-computed in the graph learning literature. In our current implementation, the space overhead is within the comparable level with the canonical self-attention. Here is a simple empirical evaluation on CoLA dataset. The JAT may increase the memory usage by 5.56%, but the efficient variant JAT$^{+}$ even requires less usage with better performance.
>
> | Memory (MB) | batch_size=32  | batch_size=64 | batch_size=128 |
> | --- | --- | --- | --- |
> | RoBERTa | 4643 | 7251 | 11091 |
> | RoBERTa-JAT | 4897 | 7351 | 12181 |
> | RoBERTa-JAT$^{+}$ | 4625 | 7225 | 11089 |
> | BERT | 4400 | 6294 | 10680 |
> | BERT-JAT | 4672 | 7330 | 12906 |
> | BERT-JAT$^{+}$ | 4360 | 6184 | 10420 |
>
> The experiment environment: Intel Xeon 3.2GHz, Nvidia V100 32GB *4, 256GB memory, Pytorch 1.10.

---

> > ### Comment · Reviewer_MTPd · 2022-08-08
> > **Update score**
> >
> > In light of the author's clarifications and willingness to perform additional experiments, I have updated my score to 6.

---

> > > ### Author Response · Authors · 2022-08-08
> > > **Response to Feedback**
> > >
> > > Dear reviewer MTPd,
> > >
> > > Thanks for your kind feedback with score updating. We will continue to improve our final version and try to include all the additional experiments ultimately.
> > >
> > > Please feel free to let us know if you have any questions or further suggestions.
> > >
> > > Yours,
> > >
> > > Authors

---

### Meta-Review · Area_Chair_w73x · 2022-08-30

**Recommendation:** Accept
**Confidence:** Less certain

**Metareview:**

The paper presents a novel architecture named jump self-attention to capture the high-order statistics in transformers. Specifically, the model builds an GCN layer on top of the attention layer, based on the attention scores. The reviews are generally positive. The major concerns are around the significance of the improvement comparing to the original softmax. The authors may want to improve this part more in the final version.

**Award:**

No

---

### Decision · Program_Chairs · 2022-09-14

Accept